# Reputation shortcoming in academic publishing

**Rémi Neveu** [ID]**[1]\*, André Neveu[2]\***

**1** affiliated to Groupe d'Analyse et de Théorie Economique (GATE LSE), Ecully, France, **2** CNRS, Université de Montpellier, Laboratoire Charles Coulomb, Montpellier, France

\* remneveu@yahoo.fr (RN); andre.neveu@umontpellier.fr (AN)

## Abstract

Editors of scientific journals make central decisions in the publication process. Information peripheral to the content of a manuscript such as the editor's professional circle and authors' publishing record may influence these decisions. This constitutes reputation whose role in the publication process remains poorly investigated. Analyzing three decades of publications of 33 *Nature* journals in physical and life sciences, we show that reputation is tied with publications at the level of editors. The establishment of a non-financial conflict of interest policy and the journals' impact factor were associated with changes in the number of publications of editors' former co-authors and authors with a publishing record in *Nature* journals. We suggest changes at the author and journal levels to mitigate the role of reputation in the publication process.

## Introduction

The publication process of scientific discoveries is built of a series of decisions made by authors, editors of scientific journals and reviewers [1,2]. Editors play a central role. They decide or decline to send out a manuscript for external review. They choose reviewers. They make the final decision to accept or reject the manuscript based on the reviewers' comments [1,2]. Given that the reputation of others influences one's own decisions [3], decisions involved in the publication process may be biased by the reputation of its actors.

Reputation is built of information peripheral to the content of a manuscript. Reputation encompasses different aspects: editors' professional circle (former co-authors and affiliations), authors' track record and prestige of a journal are some of them. The influence of reputation in the publication process is considered a bias and, under some circumstances, a non-financial conflict of interests (COI) [2,4,5]. Non-financial conflicts of interests are interests that do not engage financial transactions but could "undermine the objectivity, integrity and the value of a publication through a potential influence on the judgments and actions of authors" [4]. Scientific journals have set policies to address this issue [4–6]: some journals state explicitly as such conflicts relationships between authors and the other actors of the publication process [5] while other journals do not when it comes to the question of their own editors [6].

**Data availability statement:** All programs and data that can legally be shared are publicly available at https://dataverse.harvard.edu (https://doi.org/10.7910/DVN/O6MQB4) and, for extended data including the PubMed database, at https://entrepot.recherche.data.gouv.fr: https://doi.org/10.57745/3QL466.

**Funding:** The author(s) received no specific funding for this work.

**Competing interests:** The authors have declared that no competing interests exist.

While some claim that decisions made by professional editors are "unbiased by scientific or national prejudices of particular individuals" [7], the interaction of reputation with the publication process has not been investigated in depth [2,8] despite several matters of concerns [8–10]. Studies focused on editors [11–13], reviewers or authors [8,14–17]. However, ties between these actors have drawn little attention. Exceptions include: a gender bias that favors authors of the same gender as the reviewer [18] or the editor [16,19]; more rapid processing of manuscripts submitted by editors' former co-authors [20]; more favorable reviewers' assessments when the reviewers are suggested by the authors than chosen by the editor [21–24]; self-publication among academic editors [11,13]. The affiliation of authors [25], the track record of authors [26] or the extent of the network of authors [27] were also associated with differences in the publication outcome but the design of these studies did not allow to associate their results with authors, editors or reviewers. An author's track record was defined as the number of publications of this author within a set of journals and published over a specific period [26].

That reputation may interact with the outcome of the publication process at the editorial level is supported by several psychological mechanisms: the use of heuristics, such as reputation [3], in fast decisions [28], a characteristic featured in editorial decisions [7]; the poor sensibility of editors to problems in the publication process [29–31]; the sensitivity of journals to reputation [32]; the subjective nature of editorial criteria for publication [33].

The role of reputation in the publication process has attracted little attention. We hypothesized that authors' ties with editors are associated with the number of their publications. Analyzing three decades of publications of 33 *Nature* journals in physical and life sciences, we show that authors with less reputation or a weaker tie to the editor have substantially lower chances to publish in the editor's journal. This encompasses three aspects of authors' reputation in the eyes of the editor: a past collaboration with the editor, an affiliation to one of the editor's former research affiliations, and the author's publication track record in *Nature* journals. These differences in the publication outcome are associated with the scientific experience of the editor and amplified by variations in the impact factor (IF) of the journal. Some of these differences changed after the establishment of a non-financial COI policy by journals. We controlled for major confounding factors such as the submission behavior of authors, the academic origin of editors or the launch of a new journal. We conclude by suggesting further changes in order to mitigate the importance of reputation in the publication process.

## Methods

### Study design

The study was longitudinal and retrospective. It used the 31850051 published articles recorded in the PubMed database until December 12 2020 downloaded from https://ftp.ncbi.nlm.nih.gov/pubmed/baseline/ and the history of the journals' IF available at Clarivate until 2019. To benefit from a homogenous editorial setup across journals sampling a large range of disciplines and a large sample size, we

focused on publications of *Nature* and its 32 related journals whose name starts with "*Nature*" and which publish original articles (98002 articles). These journals include 322 professional editors and span a large range of scientific disciplines across the life and physical sciences (Table 1). The inclusion of only professional editors in the study allows to control for the confound of the research activity of academic editors. The focus on one journal family allows to control for differences in policies between families of journals. Data of the editors' biography were collected from the *Nature* website, LinkedIn and other Google searches.

These data were complemented by the history of the number of submissions of original articles to *Nature* from 1997 to 2017 available at https://www.nature.com/nature/for-authors/editorial-criteria-and-processes. These data were the only data of submissions which were publicly available. Nonetheless, we performed analyses to quantify the impact of a change in submission behaviors of authors over publications after the editor's appointment (S1 §1.10.7 in S1 File).

The study complied with the French law protecting participants to biomedical research and with the European Union rules protecting personal data. The full protocol is described in S1 §1 in S1 File. All sources of data were independent and publicly accessible except for Clarivate.

### Inclusion criteria

Journals were included in the study if they:

1- belonged to the *Springer Nature* group and were referenced in the *Nature* website (https://www.nature.com/siteindex) on November 14 2020,

2- had a name starting by "*Nature*" but not by "*Nature Reviews*"

3- published original articles in English from primary research of all countries (see S1 §1.3 in S1 File).

Primary research includes data whose results have never been published before. Editors were included in the study if they:

1- belonged to one of the journals included in the study,

2- were referenced in the *Nature* website between November 16 2020 and December 15 2020 as editors of one of the journals included in the study

3- made editorial decisions about manuscripts or comments submitted for publication by researchers (see S1 §1.3 in S1 File).

The inclusion flow chart is depicted in Fig 1. The selection biases due to the inclusion of editors was likely limited (S1 §1.10.10 and S1 §2.7.7 in S1 File).

### Data analyses

Data analyses were split into three steps: a preprocessing step, a filtering step and a statistical step. Limits of the methodological design and statistical analyses are specified in S1 §1.9.4 in S1 File.

**Preprocessing step.** The data of published articles were preprocessed in order to get two independent sets of variables (S1 §1.5.2 to §1.7, S1 Fig 3 in S1 File). In both sets of variables, we focused on original articles rather than on all articles because original articles are the main source of new discoveries. Only original articles published in *Nature* journals from 1990 to 2020 were considered in the analyses to control for missing data or misclassification before 1990 (S1 §1.5.2 in S1 File). An algorithm linking articles through their authors, authors' affiliation and key words of articles was designed to get a first disambiguation of author's name (S1 §1.6.6.1 in S1 File).

Table 1. Characteristics of all *Nature* journals included in the study. Research and editorial experiences are computed for editors who had no homonym: thus, in some journals, only one editor satisfied this criterion, yielding a standard deviation (SD) equal to 0. 'Editorial experience' is limited to the experience at *Nature* journals. Research and postdoc were experienced prior to the editors being appointed by *Nature* journals. The number of articles as first or last author refers to original articles (see supplementary method for the criteria used to define original articles).

| Journal | Year of launch | N editors included in the study | N editor with only a PhD experience (%)* | Editorial experience (years, mean (SD)) | Research experience (years, mean (SD)) | N articles as first or last author (mean (SD)) | Postdoc experience (years, mean (SD)) | Impact factor in 2019 |
|---|---|---|---|---|---|---|---|---|
| Nature | 1869 | 47 | 13 (27.7)** | 14.3 (8.47) | 7.07 (4.1) | 2.26 (2.83) | 3.12 (3.6) | 42.8 |
| Nature Aging | 2020 | 3 | 0 (0) | 4.88 (6.07) | 8.5 (2.12) | 2 (2.83) | 5 (4.24) | – |
| Nature Astronomy | 2017 | 4 | 0 (0) | 4.42 (0.12) | 10 (1.41) | 0.5 (0.707) | 7 (1.41) | 11.5 |
| Nature Biomedical Engineering | 2016 | 4 | 0 (0) | 6.54 (2.81) | 11.7 (2.08) | 3.5 (3.32) | 6 (1) | 19 |
| Nature Biotechnology | 1996 | 8 | 1 (12.5) | 16.2 (6.97) | 7.5 (3.54) | 1.8 (3.49) | 2.5 (3.54) | 36.6 |
| Nature Cancer | 2019 | 4 | 1 (25) | 4.06 (5.41) | 10.3 (5.13) | 2 (1.73) | 5 (5.57) | – |
| Nature Catalysis | 2018 | 4 | 0 (0) | 2.19 (1.61) | 7.5 (0.707) | 2.33 (2.52) | 3 (0) | 30.5 |
| Nature Cell Biology | 1999 | 5 | 1 (20) | 4.33 (0) | 8 (0) | 0 (0) | 2 (0) | 20 |
| Nature Chemical Biology | 2005 | 5 | 2 (40) | 10.5 (8.13) | 10 (5.66) | 3.5 (2.12) | 4.5 (6.36) | 12.6 |
| Nature Chemistry | 2009 | 6 | 5 (83.3) | 8.06 (6.42) | 4.33 (1.53) | 1.5 (1.73) | 0.67 (1.15) | 21.7 |
| Nature Climate Change | 2011 | 6 | 0 (0) | 3.29 (3.7) | 9 (0) | 0.5 (1) | 4 (0) | 20.9 |
| Nature Communications | 2010 | 107 | 15 (14) | 3.07 (2.39) | 8.82 (4.31) | 2.25 (2.26) | 4.27 (3.97) | 12.1 |
| Nature Computational Science | 2021 | 4 | 1 (25) | 1.59 (1.18) | 8.5 (0.707) | 1 (1.41) | 4.5 (0.707) | – |
| Nature Ecology & Evolution | 2016 | 6 | 2 (33.3) | 7.32 (5.06) | 6.25 (2.06) | 1.2 (1.64) | 1.6 (1.67) | 12.5 |
| Nature Electronics | 2018 | 4 | 2 (50) | 3.17 (0) | 4 (0) | 0 (0) | 0 (0) | 27.5 |
| Nature Energy | 2015 | 6 | 1 (16.7) | 6 (4.83) | 5 (0) | 1 (1.41) | 1 (0) | 46.5 |
| Nature Food | 2019 | 4 | 0 (0) | 1.75 (0) | 13 (0) | 1 (0) | 9 (0) | – |
| Nature Genetics | 1992 | 6 | 1 (16.7) | 9.59 (6.79) | 6 (0) | 0.67 (1.15) | 2 (0) | 27.6 |
| Nature Geoscience | 2008 | 7 | 3 (42.9) | 6.69 (4.1) | 4.67 (1.53) | 0.6 (0.894) | 0 (0) | 13.6 |
| Nature Human Behaviour | 2016 | 6 | 3 (50) | 2.82 (1.6) | 7.4 (2.3) | 3.6 (3.65) | 2.8 (3.11) | 12.3 |
| Nature Immunology | 2000 | 4 | 0 (0) | 14.1 (5.08) | 12.3 (5.77) | 1.67 (1.15) | 8 (4.36) | 20.5 |
| Nature Machine Intelligence | 2019 | 4 | 2 (50) | 8.56 (10.3) | 10.3 (8.5) | 3.33 (1.53) | 5.67 (7.37) | – |
| Nature Materials | 2002 | 8 | 0 (0) | 7.88 (6.97) | 9.4 (6.02) | 1.5 (1.64) | 5.6 (5.94) | 38.7 |
| Nature Medicine | 1995 | 9 | 0 (0) | 5.77 (6.62) | 8.33 (2.52) | 1.4 (0.894) | 3.33 (1.53) | 36.1 |
| Nature Metabolism | 2019 | 3 | 1 (33.3) | 0.63 (0.53) | 7.5 (3.54) | 3 (1.41) | 2 (2.83) | – |
| Nature Methods | 2004 | 7 | 2 (28.6) | 4.5 (3.26) | 9.67 (3.06) | 3.67 (3.79) | 5 (2.65) | 30.8 |
| Nature Microbiology | 2016 | 7 | 3 (42.9) | 1.61 (2.12) | 5.25 (1.5) | 1.25 (0.96) | 0.75 (0.96) | 15.5 |
| Nature Nanotechnology | 2006 | 6 | 1 (16.7) | 7.3 (5.14) | 9.2 (2.59) | 3.4 (2.19) | 4.8 (3.42) | 31.5 |
| Nature Neuroscience | 1998 | 7 | 0 (0) | 6.98 (3.16) | 9 (2.65) | 3.25 (4.57) | 3 (1.73) | 20.1 |
| Nature Photonics | 2007 | 4 | 2 (50) | 13.8 (1.53) | 5.67 (3.79) | 0.67 (0.58) | 1.67 (2.89) | 31.2 |
| Nature Physics | 2005 | 7 | 0 (0) | 6.59 (3.32) | 10.3 (2.52) | 0.75 (0.96) | 6.33 (3.21) | 19.3 |

*(Continued)*

**Table 1.** (Continued)

| Journal | Year of launch | N editors included in the study | N editor with only a PhD experience (%)* | Editorial experience (years, mean (SD)) | Research experience (years, mean (SD)) | N articles as first or last author (mean (SD)) | Postdoc experience (years, mean (SD)) | Impact factor in 2019 |
|---|---|---|---|---|---|---|---|---|
| Nature Plants | 2015 | 5 | 3 (60) | 12.6 (12.9) | 9.67 (8.14) | 1 (1.73) | 5 (8.66) | 13.3 |
| Nature Sustainability | 2018 | 5 | 1 (20) | 5.25 (3) | 9 (2.65) | 0.75 (0.96) | 2.25 (1.71) | 12.1 |

*we considered the 322 editors for this variable only. For the other variables, we considered the 190 editors included in the statistical analyses (Fig 1).

**8 (20.5%) when excluding all editors (n=8 out of 47) handling submissions with minimal new supporting research findings (S1 §1.3.1 in S1 File).

**Fig 1. Inclusion flow charts for journals (A) and editors of *Nature* journals (B).**

The first set of variables focused on the number of publications of each author. It aimed at addressing questions related to the editor's professional circle. For each editor, we tracked each author who published in the editor's journal during the two years before the editor's appointment at *Nature* journals or the two years after it (median number of authors: 56850 (interquartile: [5775–130417]), S1 §1.9.3.1 in S1 File). For each of these two periods, the number of the author's original articles published in the editor's journals was computed. Therefore, an article with several authors was counted once for each of its authors. Each author was then categorized:

- as whether he/she was a former co-author of the editor,

- as whether he/she was affiliated with one of the former research institutions of the editor.

Note that an author may share one of the former research affiliations of the editor without being one of his/her former co-authors: for instance, in a given university, a researcher publishing an article in neuroscience in the editor's journal may have never collaborated with the editor even though this latter has also worked at the same university and also published articles in neuroscience. Editors were also categorized as whether they were appointed before or after the establishment of a policy addressing non-financial COI in *Nature* journals in February 2018 applying to authors and editors [4]. This first set of variables was then used in the statistical analyses whose results are reported in the subheaders "Former co-authorship as a reputational input to the editor", "Former editor's research affiliation as a reputational input to the editor", "Interaction with the setup by journals of a non-financial conflicts of interest policy" and in three figures mentioned in Table 2 (i.e. Figs 2A, 3B and 3C).

We did not extend the ties to authors who never published with the editor but who were co-authors of editor's former co-authors. This extension may be viewed as an indirect, or a second level, tie with the editor.

**Table 2. General linear statistical models and mixed statistical models used in the analyses to plot the figures. The coefficients (beta) of the regressors in bold are used to plot the figures of the main text referenced in the first column. Regressors in italic are categorical variables. The random effect of all mixed models is set over the intercept.**

| Figure | Dependent variable | Regressors and interactions | Random effect |
|---|---|---|---|
| 2A | N articles of an author **after the editor's appointment** - N articles of an author **before the editor's appointment** | • Number of articles published by the author during the 2 years before the editor's appointment<br>• *Type of author (editor's former co-author/other authors)* | Editor |
| 2B | % of original articles by new authors only each year | • Type of journal (general/specialized)<br>• Number of years since 2021 (i.e., -1, -2, -3…)<br>• Each editor's characteristics (time spent in academia, time spent as editor, number of editor's original articles published as first or last author) mentioned in Fig 1B<br>• Interaction between years since 2021 and journal type<br>• **Interactions between years since 2021 and each editor's characteristics** | Editor |
| 2C (orange and red lines) | % of original articles by only new authors each year | • Type of journal (general/specialized)<br>• **Year of publication after the launch of the journal**<br>• **Interaction between year of publication and journal type** | Journal |
| 2C (blue and light blue lines) | % of original articles by known intermediate authors each year | • Type of journal (general/specialized)<br>• **Year of publication after the launch of the journal**<br>• **Interaction between year of publication and journal type** | Journal |
| 3A (the two bars on the left) | N articles by a given type of authors at year n+1 − N articles by a given type of authors at year n | • Type of journal (general/specialized)<br>• Type of article (first and last author new/first or last author with a track record in Nature journals)<br>• **Impact factor year n – impact factor year n-1**<br>• **All two-way and three-way interactions** | Journal |
| 3A (the two bars on the left) | N articles by a given type of authors at year n+2 − N articles by a given type of authors at year n+1 | • Type of journal (general/specialized)<br>• Type of article (first and last author new/first or last author with a track record in Nature journals)<br>• **Impact factor year n – impact factor year n-1**<br>• **All two-way and three-way interactions** | Journal |
| 3B | N articles of an author **after the editor's appointment** - N articles of an author **before the editor's appointment** | • Number of articles published by the author during the 2 years before the editor's appointment<br>• *Type of author (editor's former co-author/other authors)*<br>• *Period of the editor's appointment (before/after the setup of the non-financial **COI** policy at Nature journals)*<br>• *Interaction between the two categorical regressors* | Editor |

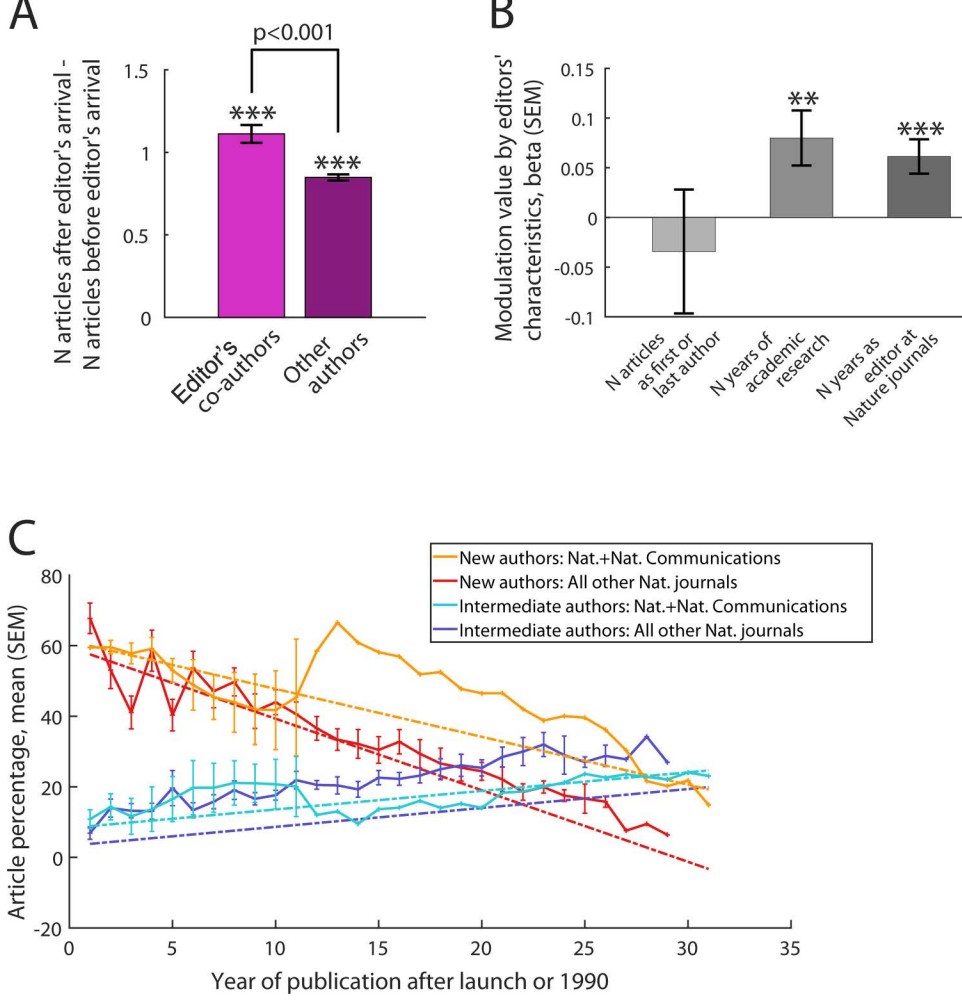

**Fig 2. Average difference in the number of articles published by each author in the editor's journal during the two years before and the two years after the editor's appointment at *Nature* journals as a function of the status of the author (A); Modulation value of the orange and red slopes of Fig 1C (computed for articles of new authors) as a function of editor's research experience and editorial experience at Nature journals (B); yearly percentage of original articles published by new authors only (orange and red) or by known intermediate authors (light and dark blue) in *Nature* journals as a function of year after journal launch (C, slope (standard error of the mean (SEM)): -1.34 (0.29), p<0.00001 (orange); -1.98 (0.15), p<0.00001 (red); 0.52 (0.16), p =0.001 (light blue); 0.76 (0.08), p<0.00001 (dark blue)).** In Fig 2B, the modulation captures the amount which must be added (if the modulation is positive), or subtracted, to the slope of Fig 2C in order to get the slope value for each editor. This amount is equal to the coefficient depicted in Fig 2C (e.g. the medium grey bar) multiplied by the number of years of academic research of the editor (for the medium grey bar). *Comparisons of bars against zero: *: p<0.05; **: p<0.01; ***: p<0.001. Data were truncated to January 1st 1990 for Nature.*

The second set of variables focused on the number of articles published per year in each journal. It aimed at addressing questions related to authors' track records and the relationship with the journal's IF. Each article was categorized as whether all its authors had no track record in *Nature* journals (i.e., articles with only new authors) and whether at least one of its intermediate authors had a track record in *Nature* journals (i.e., articles with a known intermediate author). Note that an article cannot be simultaneously in the two categories. We then computed for each year of publication of the journal:

- the rate of original articles with only new authors

- the rate of original articles with a known intermediate author.

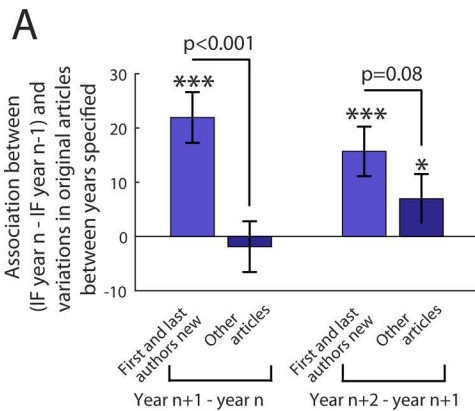

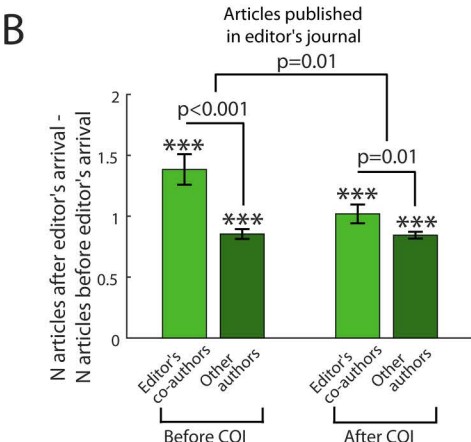

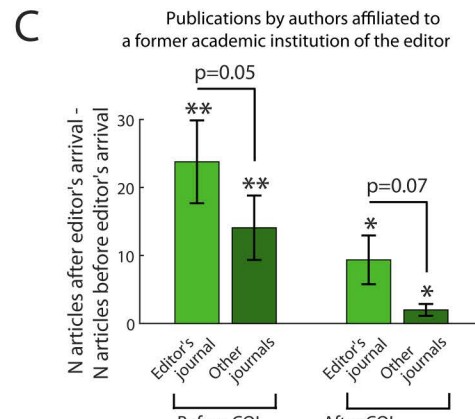

**Fig 3. Linear association (i.e., beta) between the variations in impact factor (i.e. $IF_{year\,n}-IF_{year\,n-1}$) and subsequent variations in articles published (i.e. $N_{original\,articles\,year\,n\,+1}- N_{original\,articles\,year\,n}$ (respectively $N_{original\,articles\,year\,n+2}- N_{original\,articles\,year\,n+1}$ for the two bars on the right)) in *Nature* and *Nature Communications* (A); Average difference in the number of articles published by authors in the editor's journal during the two years before and the two years after the editor's appointment at *Nature* journals as a function of the status of authors and when the editor was appointed (i.e., before or after the establishment of the non-financial conflict of interest (COI) policy, B); Same as Fig 3B except that the number of articles published by editors' former co-authors is replaced by the total number of articles by authors affiliated to research institutions where the editor worked before joining *Nature* journals (C).** *Comparisons of bars against zero: \*: p<0.05; \*\*: p<0.01; \*\*\*: p<0.001.*

This second set of variables was used in the statistical analyses whose results are reported in the subheaders "Author's track record as a reputational input to the editor", "Interaction with the reputation of the journal" and in three figures of Table 2 (i.e. Figs 2B, 2C and 3A).

The track record of authors was computed in *Nature* journals rather than in all journals because: homonyms are more likely when considering a higher number of journals; the editor's memory of authors' names is more likely to impact fast editorial decisions [7] than a time consuming search of the full track record of each author of an article [28]. For example, authors' names could be heard during weekly editorial meetings held at this type of journals [34] and the *Nature* publishing group expects their editors to liaise with researchers and editors of other *Nature* journals as stated in advertisements for editorial jobs (S1 §2.2.6 in S1 File).

For consistency with the duration used to compute the IF, a new author was defined as an author who had not published in any *Nature* journal for two consecutive years at least. We studied how authors' ranking as first, last or intermediate interacted with the results (see S1 §2.6.1 in S1 File).

**Filtering step to remove outliers.** The first set of variables was filtered to address homonym issues (S1 §1.6.6 in S1 File). This filtering relied on an algorithm generalizing the standard method to remove outliers (S1 §1.6.6.2 in S1 File). This method is based on the detection of data below average-n*standard deviation or above average-n*standard deviation with n being equal typically to 2, 2.5 or 3.

In the second set of variables, we removed original articles from authors whose name was referenced in more than 50 articles in a given *Nature* journal over its entire duration of publication (S1 §1.6.7 in S1 File).

The efficiency of this filtering (S1 §1.6.6.2 in S1 File) and its effect over the results of the statistical analyses were systematically quantified (S1 §1.11 in S1 File).

**Statistical analyses.** Quantitative data were analyzed with Student t-tests, Wilcoxon or the Mann-Whitney tests whenever appropriate, and qualitative data with Fisher exact or chi square tests whenever appropriate. Most analyses relied on general linear and mixed models (Table 2, S1 §1.9 and §1.10 in S1 File). We report here the main models. The others are described in S1 §1.9 and §1.10 in S1 File. Two types of models were designed: one for the results related to the editor's professional circle and the second one for the rate of articles by new authors or known intermediate authors.

We focused on three variables which captured ties between authors and editors: the number of publications of editor's former co-authors in the editor's journal, the number of articles published in the editor's journal whose authors were affiliated with one of the editor's former research affiliations, the track record of authors in the editor's journal family. Thus, authors have a weaker tie to the editor when they have not worked with the editor previously, or when they have a different affiliation than the former editor's ones, or when they have a lower track record in the editor's journal family.

These three variables may uncover several confounds unrelated to reputation or to editors: the submission behavior of authors, the academic origins of editors, the content of articles are some of them. We tackled these issues by controlling them with the study design, by estimating the biases of these confounds over our results and by strengthening in three ways the tie between the editor and the three variables. First, for publications of editor's former co-authors and authors affiliated to one of the editor's former affiliations, we investigated differences between the periods before and after the editor's appointment at his/her journal. We also quantified the interaction with the setup, by the editor's journal, of a non-financial COI policy addressing ties between authors and editors [35–37]. Second, because the impact factor (IF) of journals may influence editors, we investigated whether the variations in the IF of the editor's journal were associated with the publications of authors with different track records in the family of journals of the editor. Third, we tested whether the observed results were associated with the scientific or academic experience of the editor.

***Former co-authorship as a reputational input to the editor.*** The first type of models was used in this analysis. We focused on the difference in the number of publications of authors after the editor's appointment as compared to before (Table 2). Analyzing this difference allows to better control for the quality of articles associated to a given author as compared to previous analyses [11] which focused on each period of publication separately. We included only editors (n=56

(36%), S1 §1.9.3.1 in S1 File) for whom at least a former co-author published in the editor's journal either during the two years before the editor's appointment or the two years after.

The difference in publications between the two periods was modeled as a linear function of the type of authors (i.e., editor's former co-authors/other authors). The level of the academic origins of editors was controlled for by adjusting the model over the track record of authors in the editor's journal (Table 2, S1 §1.9 in S1 File). Editors were set as a random factor over the intercept. In a second step, the editor's experience was added in that model to test for an interaction with the type of author (S1 §1.9 in S1 File): this interaction allowed to strengthen the tie with the editors in the results.

Then a mixed effect logistic regression was used to quantify the chances of editor's former co-authors in succeeding in publishing after the editor's appointment. The number of publications of each author after the editor's appointment was converted into a binary variable: 1 for authors who had at least one article, 0 otherwise. This binary variable was modeled as a logit function of the type of authors (i.e., editor's former co-authors/other authors) and the number of articles published by the author before the editor's appointment. Editors were set as a random factor over the intercept. This model allowed to calibrate our results by comparing the chances to publish associated to the tie between authors and editors with the chances to publish associated to the track record of authors irrespective of their tie with the editor.

***Former editor's research affiliation as a reputational input to the editor.*** For each editor, we computed the difference, between before and after the editor's appointment, in the number of articles published in the editor's journal and in which at least an author was affiliated with one of the editor's former research institutions. The same difference was computed for all journals included in the study except the editor's one. These two numbers were compared with a paired sample Student t-test.

***Author's track record as a reputational input to the editor and interaction with the reputation of the journal.*** The second type of models focused on the number of original articles by new authors published each year in each journal (Table 2). In models of Fig 2B and 2C, we computed for each year of publication the rate of original articles published by new authors (Table 2). This rate was modeled as a function of time (i.e., the year). In the model of Fig 3A, we computed the difference, between year n+1 and year n, in the number of articles published by a type of author (i.e., first and last authors with no track record in *Nature* journals vs first or last author with such a track record) over the year (Table 2). This difference was then modeled as a function of the difference between year n and year n-1 of the impact factor of the journal, the type of author and their interaction. The model was re-estimated using the difference between year n+2 and year n+1 instead of the difference between year n+1 and year n. To account for the heterogeneity among journals, we distinguished between generalist and specialized journals in these models (Table 2).

***Interaction with the setup by journals of a non-financial conflicts of interest policy***. The first type of models was used again in this analysis. We complemented the linear mixed model involved in the analysis of the difference in publications by editor's former co-authors after the editor's appointment as compared to before (Table 2). The use of a linear mixed model allows to account for the unbalanced nature of the data: the number of editor's former co-authors differs between editors and the number of publications differs between the former co-authors of a given editor. The same considerations apply for the other authors over the periods of interest around the editor's appointment date. We discriminated when the editor was appointed: before or after the setup of the non-financial COI policy by *Nature* journals [4]. An interaction between the period of recruitment of the editor (before/after the setup of the non-financial COI policy) and the type of author (editor's former co-author/other authors) was added. Results are reported in Fig 3B.

***Additional control analyses***. All these analyses were complemented by other linear mixed models and logistic regressions in order to control that our results were not biased by:

- the quantification of a potential effect of the submission behavior of authors over publications (S1 §1.10.7 in S1 File) to complement our results obtained with submission data,

- the quantification of potential biases due to the selection of editors (S1 §1.10.9 and §1.10.10 in S1 File),

- the launch of a new journal (S1 §1.10.2 in S1 File),

- the labeling of articles as original research, the author ranking rule (S1 §1.11.1 in S1 File),

- the time between the editor's appointment date and the date of publication of his/her first article handled (set to 8 months, S1 §1.6.3 in S1 File) which included the duration of the publication process and the duration of the editorial training provided by the journal (S1 §2.4.4 in S1 File),

- the role of the handling editor (S1 §2.7.3 in S1 File).

Finally robustness analyses were run to check that the definition of new authors did not bias our results (S1 §1.11 and §2.3 in S1 File).

## Control studies

Two ancillary studies were run to control for the contribution of the recruitment policy of editors and the content of the published articles to our results (S1 §1.1 in S1 File).

## Results

### Characteristics of editors

We included in our analyses 190 *Nature* journal editors (Fig 1), who had the typical profile of more junior scientists: they had an average of 4.74 years standard deviation (SD)=3.67) of academic research experience after the doctorate before leaving academia, which is lower than the previously reported 20 years of research experience of academic editors of other journals at the start of their editorship [11]. Additionally, we could not identify any review article nor any meta-analysis that yielded a first or last authorship for the editors we included in our analyses. These editors published substantially fewer articles as last author than as first author (mean (SD): 0.24 (0.77) vs 1.76 (2.11), p<0.00001). For the editors of *Nature*, the flagship journal, 20.5% had no academic research experience after the doctorate that we could identify. A similar proportion applied to the other *Nature* journals (mean (SD): 22.8% (22.3%), p=0.23, Table 1). These editors were recruited among research groups who published more articles in *Nature* journals than the other research groups (S1 §2.2.4.4 in S1 File).

We identified co-authors for editors in 82% of the 190 editors of *Nature* journals included in the analyses. That we had editors for whom we could not find co-authors may be the consequence of a change in the name of the editor after the end of the editor's research career. It may typically occur when women get married. Another reason for having editors with no co-author identified may be a research topic of the editor which is not referenced in PubMed. Editors had a median number of 24 co-authors (interquartile: [9–44] co-authors) which is close to recent reported results among scientists (i.e., 29) but lower than the 163 collaborators of academic editors at the start of their editorship [11]. Only a subpart of these co-authors published in the editor's journal during the two years before or the two years after the editor's appointment (mean (SD): 4.4% (8.3%)).

### Former co-authorship as a reputational input to the editor

We investigated how a manuscript of a former editor's co-author was assessed compared to other publications. Co-authors published articles with the editor mainly outside *Nature* journals before the editor's appointment at his/her journal. Among authors with the same publishing track record in the editor's journal before the editor's appointment, all authors published more original articles within the two years following the editor's appointment, compared to the two years before (Fig 2A). Moreover, the average increase in publications of editor's co-authors after the editor's appointment was 31% higher than the one of the other authors (Fig 2A). This difference with the other authors represented 120% more chances

for co-authors of the editor to publish in his/her journal after his/her appointment than the other authors (odds ratio [95% confidence interval]: 2.2 [1.5–3.24], p<0.0001). To calibrate this effect, we quantified the chances of subsequent publications for authors with an established track record of publication in the editor's journal irrespective of whether they were former co-authors of the editor. For each additional original article published in the two years before the editor's appointment, these authors had 161% more chances to publish another original article in this journal after the editor's appointment (odds ratio [95% confidence interval]: 2.6 [2.59–2.62], p<0.0001, supp. §2.1.1). The increase in author's chances to publish an original article in the editor's journal was therefore similar when an author: was a former co-author of the editor on one side, and, on the other side, had one more article in his/her track record in the journal of the editor. Additionally, the time spent by an editor in academia was associated with the increase in publications of editor's co-authors but not with those of other authors (S1 §2.1.2 in S1 File). It thus establishes a relationship between this increase and editors.

A change in the submission behavior of editor's former co-authors after the editor's appointment as compared to before can only contribute marginally to the higher increase in their publications depicted in Fig 2A. Indeed, in the editor's journal, the percentage of editor's former co-authors who published only before the editor's appointment was similar to the percentage of those who published only after (S1 §1.10.7 and §2.7.5 in S1 File). Similar results were obtained when comparing the percentage of editors whose former co-authors published only before the editor's appointment to those whose co-authors published only after (S1 §1.10.7 and §2.7.5 in S1 File).

### Former editor's research affiliation as a reputational input to the editor

Institutions where authors are affiliated is another element potentially influencing the editor when assessing a manuscript. Institutions where the editor worked previously as a researcher contribute to the reputation of authors of these institutions to the editor even if these authors are not former co-authors of the editor. In the two years following an editor's appointment, 29.6 more original articles were published in the editor's journal with at least one author from a research institution where an editor was previously affiliated, compared to the prior two years. Such an increase was not present in the other *Nature* journals (S1 §2.1.4 in S1 File) and is therefore unlikely to be attributed to a concomitant improvement in the scientific output of these institutions.

### Author's track record as a reputational input to the editor

Considerations independent from the editors' professional circle might also influence editors. Whether the editor has already seen, during his or her editorial career, the author's name in another article previously published in *Nature* journals is another way for the editor to know the author. This tie is weaker than the one between editors and their former co-authors but impacts more authors. Authors with a publication record in these journals are therefore "known" to the editor as compared to "new" (S1 §1.5 in S1 File) at the time of the assessment of a new manuscript. The percentage of original articles published by only new authors in *Nature* journals decreased over the years (Fig 2B and 2C). This decrease was larger for editors with a shorter time spent in academia or as an editor at *Nature* journals (Fig 1B). The percentage of original articles published by known authors that were in middle authorship position increased over the years (Fig 2C). This increase was independent of any of the editors' characteristics captured in Fig 2B (p=0.97, p=0.68 and p=0.15 respectively). These results establish an association between editors and the decrease in the publications of new authors.

### Interaction with the reputation of the journal

The impact factor (IF) is associated with the prestige of journals (S1 §2.2.2 in S1 File). We assessed how the temporal variations of a journal's IF impacted subsequent publications in this journal. During the years following a decrease in IF, this decrease was associated with fewer publications of authors without a publication record in *Nature* journals as compared to the authors having such a record (Fig 3A, supp. §2.1.6). This effect has been increasing since 1997 (S1 §2.1.6 in S1 File). However, there was no association between IF variations and the total number of submitted articles (supp.

§2.1.6). Consistently with our results on the publications of editors' former co-authors, our results suggest that the submission behavior of authors has little chance to contribute to the difference in publications between new and known authors.

**Interaction with the setup by journals of a non-financial conflicts of interest policy**

Changes in the publication process may be introduced to reduce the role of reputation of authors and journals in the publication process. As an example of such a change, we quantified the modifications in the publication outcome after the establishment of a policy addressing non-financial COI in *Nature* journals in February 2018 applying to authors and editors [4]. We focused on publications in the editor's journal and on publications of authors affiliated with one of the editor's former research institution (S1 §1.9.3.6 and §1.9.3.7 in S1 File). Among editors appointed before February 2018, their former co-authors published more in the editor's journal after the editor's appointment as compared to before (Fig 3B). Likewise in Fig 2A, this increase was larger than for the other authors. This difference between editors' co-authors and the other authors was reduced after February 2018. After February 2018, as compared to before, there was a smaller increase in the number of original articles published by editors' former co-authors (p=0.01) while there was no change for the other authors (p=0.85). In the editor's journal and the other *Nature* journals, there was no difference after February 2018, as compared to before, for the increase in the number of publications of authors affiliated to one of the former research institutions of the editor (p=0.17, Fig 3C). Favoring former co-authors may be regarded as a COI [4,5]. These results suggest that non-financial COI may underlie the increase in the publications by editors' former co-authors. Requesting from editors to liaise extensively with researchers outside the publication process (S1 §2.2.6 in S1 File) may create biases or non-financial COI. Our results also strengthen the suggestion that reputation interacts with the publication process at the level of editors.

## Discussion

We showed consistent results for authors who were a former co-author of the editor, authors affiliated with a former research institution of the editor, and authors who had a larger track record in the family of journals of the editor. After the editor's appointment as compared to before, these authors had more publications in the editor's journal than the other authors. These results were associated with the scientific experience of the editors. Additionally, variations in the impact factor of the editor's journal and the setup, by the journals, of a non-financial COI policy to address ties between actors of the publication process interacted with the results. Altogether, these results suggest interactions at the level of editors between the reputation of authors and the outcome of the publication process.

Our results are consistent with psychological biases in decisions: reputation of others [3] or a junior experience [38] interact with decisions. Junior experience is illustrated by the lower scientific experience of professional editors as compared to academic ones [11]. Our results are also consistent with literature reporting biases in the publication process: a shorter duration of the publication process observed for editor's former co-authors [20], a higher proportion of articles published by authors of the same gender as the editor [16,19], self-publication behavior among academic editors [11,13]. We also reproduced previously published results [25,26,39] and extended them by linking them to the editors.

Our results have direct implications: they do not support the claims that professional editors "make decisions unbiased by scientific or national prejudices of particular individuals" [7]. They highlight a confound that is usually not taken into account in studies drawing conclusions on authors based on their publication data [39]. Our results question also the fairness of the publication system currently put forward by journals [7,19,40,41].

The observational nature of our study sets limitations in the identification of the causal role of authors, editors and reviewers in our results. Nonetheless, our study includes the results of an intervention set by journals and editors [4,5] which focused on the relationship between authors and editors. Changes in the origins of authors after the introduction of this policy support the idea that the reputation of authors interacts with the publication process.

The submission behavior of authors can only contribute marginally to our results: a similar proportion of editor's co-authors publish exclusively before the editor' appointment or exclusively after; results obtained with publication data were

not observed with submission data; all our main results related to the publications of editor's co-authors and new authors were associated with editor's characteristics; published articles provide a more homogenous dataset than submissions in terms of quality allowing thus to better control for this confounding factor.

Like for the submission behavior of authors, the contribution to our results of a tie between authors and reviewers [21–24] is likely limited: as compared to other authors, the increase in chances to publish is substantially larger for editor's former co-authors than for authors with a tie to the reviewers (120% vs 85% [24]); the variations in authors' publications correlated with the editors' characteristics, were synchronized with the editor's appointment at the journal and interacted with the journal IF and a non-financial COI policy set by the journal; we compared two periods of publications within the same journal centered on the editor's appointment and the periods before and after the setup of a non-financial COI policy.

We did not control for the gender of authors and editors. Nonetheless, the increase in chances to publish associated to editor's former co-authors is substantially larger than the increase associated to the gender of authors (120% vs 23% [18]). The contribution to our results of the gender of authors and editors is therefore limited.

Because research topics in physical sciences are only partially referenced in the PubMed database [42], the publication track record of editors of *Nature* journals in those fields may be underestimated. To circumvent that, we also collected the time spent in academia by editors. Results observed with the publication track record were in agreement with those observed with the editor's time spent in academia.

The magnitude of some differences reported in our results and the proportion of one third of editors who had former co-authors publishing in their journal may be perceived as limited. Nonetheless, the stronger competition to publish in *Nature* journals than in other journals (around 8% of submissions at *Nature* are accepted for publication [7]) likely increases the impact of a difference even objectively small in the chances to publish in such journals. Moreover, we report differences in chances to publish that are higher than the ones associated to a gender effect [18] or observed in ties between authors and reviewers [24]. Additionally, the unavailability of the full submission data does not allow addressing whether editors who had no former co-author publishing in their journal were exposed to submissions by their former co-authors.

Other parameters may contribute to our results (S1 §1.9.4 in S1 File). The launch of a new journal, the reputation of reviewers, a potential pressure of editor's former co-authors after the editor's appointment, the author's position in the academic research institution are some of them. Author's position in the academic research institution is partially captured by the authorship rank in the article: authors with an established position are more likely to be last author than authors with a post-doc position who are likely to be first author. Nonetheless results reported in Fig 2C, additional analyses and previous results [43] suggest that several confounding factors such as the launch of a new journal, the scientific content of manuscripts, the author's position in an academic research institution or the recruitment of new editors do not contribute to decrease the differences in publication records between authors with different ties to the editor (S1 §2.2.3 to §2.2.5 in S1 File).

Our results open directions for future studies. Editors know the names of the reviewers [2]. One may wonder whether reviewers' reputation plays a role in the publication process in the same way that authors' reputation may. This would complement previous results about ties between authors and reviewers [21–24]. Potential differences may be found between research fields or type of publishers (for-profit vs the others). Such shortcomings in the publication process may also explain the Matthew effect of accumulated advantage in science [44,45].

Addressing publication shortcomings is needed to support the fairness of the publication system [19,40,41]. The likelihood of obtaining funding, or securing appointments or promotions, increases when authors publish papers in influential journals [10,44–47]. Articles published in such journals can shape decisions in academia as well as in the larger social and political context [2,48,49].

As previously argued [50], our results suggest that regulations in the editorial policies may have chances to mitigate the role of the editors' or authors' professional circle in the publication process. At their appointment, editors would report to their journal the names of all their former affiliations and former co-authors. Editors would then not be able to handle

manuscripts from any of their co-authors or any author affiliated with one of their former research institutions for a specific duration. The name of the handling editor could be systematically disclosed in the article to improve transparency [13]. This disclosure may be set within the framework of an open peer review model of publication in which reviewers, editors and authors' names are disclosed to one another [51,52]. Co-authors of the editor may be invited to refrain from submitting more articles in the editor's journal for a specific duration after the editor's appointment. Such embargoes already exist for reviewing activities in the policies of grant agencies. Direct contacts between editors handling manuscripts and researchers need to be limited to the publication process.

However such regulations do not address a role of an author's track record or the journal's impact factor in the publication process. As in clinical trials in which blindness of all actors is the gold standard to control for biases in any type of assessment involving human beings, anonymizing the manuscript may mitigate the role of the author's track record, the journal's IF and the editors' or authors' professional circle. Masking of authors' names to reviewers is a submission option in journals [1,53]. Extending the masking of authors' names and affiliations to editors and making it systematic may better address reputation biases [46,54]. Such a masking, so called triple blinding [53], is already implemented in a limited number of journals in philosophy and quality improvement [53]. It is not straightforward and requires substantial adaptations from authors and journals. Authors would need to remove elements of their manuscript and cover letter that would allow to identify its origin [53]. This requires from authors to mention neutrally their previous work [53]. Elements that could not be removed (e.g., names and funding information) would be placed in a specific section of the manuscript that could be automatically masked by the submission system [53]. It would be disclosed only after the final editorial decision. Editorial assistants would assess the adequate anonymization of the manuscript before editors would have access to it. Artificial intelligence implemented in the submission system might help in this process.

The triple blinding and open peer review models may be complementary: the blinding may be set until the final editorial decision and the names and affiliations of the handling editor, the reviewers and authors would be disclosed to one another after the final editorial decision whatever this latter is. One may expect that editors, authors and reviewers may thus be more careful about conflicts of interests in their assessment of the work of the other actors of the publication process [52].

In conclusion, ties between authors and professional editors, such as a past collaboration, interact with the outcome of the publication process at the level of editors. The setup of a non-financial conflicts of interests policy seems to mitigate this interaction. That journals clarify on their website the situations they consider as non-financial conflicts of interests and homogenize this list between journals would contribute to strengthen the credibility of the featured fairness of the publication process [7,19,40,41]. The interaction between publications and ties between reviewers and editors is beyond the scope of this study. Our results support the setup of a systematic triple blinding in the publication process which may be complemented by an open review process. Changes addressing reputation biases will contribute to improve the fairness of and the public trust in the publication system [48].

## Supporting information

**S1 File. The file "S1 Supplementary materials and figures" contains complementary information about the method, additional results, tables and figures.** References with § in the main text refer to the corresponding paragraph of this file.
(DOCX)

## Acknowledgments

We thank Guillaume Barbalat, MD, PhD, Joao Bechelli Azzi, PhD, Amaury Gaussen, PhD, Mateus Joffily, PhD, Karine Joubert, MSc, Georges Lutfalla, PhD, Dorine Neveu, PhD for their useful considerations and comments on the manuscript. We thank Matthew Maxwell Burton, MSc, for reviewing the English of the manuscript. We warmly thank Hidde

Ploegh, PhD, for very helpful editorial suggestions. The computing center of the CNRS IN2P3 (Institut National de Physique Nucléaire et de Physique des Particules) and Sylvain Maurin provided grid computing facilities and its support.

## Author contributions

**Conceptualization:** Rémi Neveu, André Neveu.

**Data curation:** Rémi Neveu.

**Formal analysis:** Rémi Neveu.

**Investigation:** Rémi Neveu.

**Methodology:** Rémi Neveu, André Neveu.

**Project administration:** Rémi Neveu.

**Resources:** Rémi Neveu.

**Software:** Rémi Neveu.

**Supervision:** Rémi Neveu, André Neveu.

**Validation:** Rémi Neveu, André Neveu.

**Visualization:** Rémi Neveu, André Neveu.

**Writing – original draft:** Rémi Neveu.

**Writing – review & editing:** Rémi Neveu, André Neveu.

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
