## [Decision Letter · Decision Letter 0]

13 Feb 2024

PONE-D-23-31262

Reputation shortcoming in academic publishing

PLOS ONE

Dear Dr. Neveu,

Thank you for submitting your manuscript to PLOS ONE. After careful consideration, we have decided that your manuscript does not meet our criteria for publication and must therefore be rejected.

I am sorry that we cannot be more positive on this occasion, but hope that you appreciate the reasons for this decision.

Kind regards,

Samuele Ceruti

Academic Editor

PLOS ONE

Additional Editor Comments:

The majors recognised by the reviewers are in such number and of such weight that they do not suggest me that the revision should be continued. I suggest the Authors thoroughly revise the paper by incorporating all reviewers' suggestions and making the entire paper much more scientific and rigorous in methodology, results and speculation.

Reviewers' comments:

Reviewer's Responses to Questions

**Comments to the Author**

1. Is the manuscript technically sound, and do the data support the conclusions?

Reviewer #1: Yes

Reviewer #2: Partly

2. Has the statistical analysis been performed appropriately and rigorously? 

Reviewer #1: Yes

Reviewer #2: I Don't Know

3. Have the authors made all data underlying the findings in their manuscript fully available?

Reviewer #1: No

Reviewer #2: Yes

4. Is the manuscript presented in an intelligible fashion and written in standard English?

Reviewer #1: Yes

Reviewer #2: No

5. Review Comments to the Author

Reviewer #1: The authors study the bias in editors in making decisions in the publication process. This bias is related to the authors' publications and the editor’s professional circle.

Also, this manuscript shows that authors with a lesser reputation or a weaker tie to the editor have a lower chance of publication. Even this editor’s odd behavior is bigger between different journals with different impact factors.

The study is very rigorous in general terms. However, the structure of the article must be modified. Some of the material proposed as supplementary should be included in the main text, and such extensive supplementary material should not be provided. On the other hand, the sections of the manuscript should be Introduction, Literary Review, and Methodology, which includes the starting hypotheses and their subsequent verification in a Discussion section. Finally, a conclusion section should be created where the most exciting aspects of the article and its weaknesses are pointed out.

Comments.-

1.-The authors must identify the Introduction section after the Abstract.

2.-I’d like the authors to explain with more detail “weaker tie to the editor”.

3.-In the study carried out in the section "Former co-authorship as a reputational input to the editor," it is unclear to me if the authors took into account authors who had not been co-authors of the editor and were not co-authored by any editor's co-author. In this case, the editor's influence could be extended indirectly. In this section, the first time in which the authors present statistics results will be needed to indicate the type of test used.

4.-In the section "Author’s track record as a reputational input to the editor," why did the authors limit the study to new authors publishing in Nature journals? Usually, a scientist has access to the different publications of an author.

5.-The period to recruit editors into the studio could be a lot longer. Have the authors looked at other time windows to recruit editors?

6.-In the box bottom in Figure 1 (in supplementary material) appears “(PhD or”. I recommend that the authors translate this figure into the manuscript.

7.-In the preprocessing step of the published papers, how do the authors get the original papers?

Reviewer #2: Thank you to the authors to submit this manuscript for review. The authors addressed the concern of publication bias based on editors' and authors' reputation. To investigate this, they reviewed three decades of publications of 33 Nature journals in physical and life sciences and they analyzed associations between publications and ties between editors and authors.

The structure of the article is very weird and accordingly very difficult to read and to understand. The first abstract is followed by a bigger abstract (?), not being clearly an introduction. This "bigger abstract" ist followed by results, than discussion and finally the method section. The authors should organize the manuscript according to the journal guidelines (https://journals.plos.org/plosone/s/submission-guidelines). Additionally, it would be recommend to use standard checklist to report research data (https://www.equator-network.org). This manuscript seems the synopsis of a PhD project (reported as a supplemental, which with 144 pages, it is of course not possible to review in this context). However, a scientific article it is not the same. The authors should focus on the problem, develope an hypothesis and test it with an appropriate sicentific method. Limitations to this should be reported in the discussion (which are missing).

1. Problem (and research question)

The authors reported the concern of publication bias based on editors' and authors' reputation. What do we already know (introduction)? The authors already reported apparently associations between gender of the editor and the authors (which is not described), so as faster submission process by co-authors of the editor. These concerns should be addressed in detail in the introduction. So, what is your hypothesis? If the hypothesis related to publication success, the analysis should consider rejected manuscript, which are here apparently non-available (?). If so, how planned the authors to go through without this information? Other problems should be considered: e.g. If an author has already published numerous articles across various journals, their expertise alone would inherently enhance the likelihood of a successful publication, irrespective of the editor's influence.

2. Methods & results

Since the hypothesis is not precisely defined, also the method section is difficult to understand. It seems that the authors compared the number of published articles in the two years before or after a submission (with or without ties to the editor). This difference seems to be very little (ca. 1.1 vs 0.9 according to figure 1 A) and also if statistically significant, is it really the "proof" of a potential publication bias? The methods section whould be "aligned" with the results sections, i.e. e.g. in each subheader of the results section should we find a subheader in the methods section, which explain the method used.

3. Results

The authors should report the "demographics": how many articles they analyzed? How many editors (this is there)? How many authors? The absolute number of article in difference should be given additionally to the proportions.

4. Discussion

The authors should focus first on their results. A limitation section is mandatory. Bias on the own results is not addressed. A comparison with other article focussing on the same problem is not given. Double blinding between authors and reviewers is increasing in the last years. The authos should address this in the discussion (not just name it).

I finally appreciate the effort of the authors to increase knowledge in the filed of the publication process. An extention of the blinding to the editors is an interesting idea. However, the authors should me more precise in the structure of the manuscript, so as to give a clear hypothesis with an appropriate design and reporting.

6. PLOS authors have the option to publish the peer review history of their article (what does this mean? ). If published, this will include your full peer review and any attached files.

**Do you want your identity to be public for this peer review?** For information about this choice, including consent withdrawal, please see our Privacy Policy .

Reviewer #1: No

Reviewer #2: No

- - - - -

---

## [Author Response · Author response to Decision Letter 1]

29 Mar 2024

Please see the submitted pdf containing the point by point answer to the reviewers.

---

## [Decision Letter · Decision Letter 1]

1 Nov 2024

PONE-D-23-31262R1Reputation shortcoming in academic publishingPLOS ONE

Dear Dr. Neveu,

Thank you for submitting your manuscript to PLOS ONE. After careful consideration, we feel that it has merit but does not fully meet PLOS ONE’s publication criteria as it currently stands. Therefore, we invite you to submit a revised version of the manuscript that addresses the points raised during the review process.

We look forward to receiving your revised manuscript.

Kind regards,

Shimpei Miyamoto

Academic Editor

PLOS ONE

Journal Requirements:

4. We note that you have referenced (van Lent M, Overbeke J, Out HJ. Role of editorial and peer review processes in publication bias: analysis of drug trials submitted to eight medical journals. PLoS One. 2014;9(8):e104846. doi: 10.1371/journal.pone.0104846. PubMed PMID: 25118182; PubMed Central PMCID: PMCPMC4130599.) which has currently not yet been accepted for publication. Please remove this from your References and amend this to state in the body of your manuscript: (ie “Bewick et al. [Unpublished]”) as detailed online in our guide for authors

Additional Editor Comments (if provided):

Journal office requests: 

* Please update the title to meet PLOS ONE submission guidelines

* On Page 16 there is extended discussion of "blinding", but this should be revised to "masking" or "anonymised", as appropriate

* The authors indicate that the journals needed to belong to the "Nature Springer" group, but it should be designated "Springer Nature"

* The authors' search is limited to Pubmed, and that presents an issue when looking at the publication record of editors working on journals like 'Nature Physics'. This should be discussed as a limitation (it is currently only touched on briefly in the Results section)

Reviewers' comments:

Reviewer's Responses to Questions

**Comments to the Author**

1. If the authors have adequately addressed your comments raised in a previous round of review and you feel that this manuscript is now acceptable for publication, you may indicate that here to bypass the “Comments to the Author” section, enter your conflict of interest statement in the “Confidential to Editor” section, and submit your "Accept" recommendation.

Reviewer #1: All comments have been addressed

Reviewer #2: All comments have been addressed

2. Is the manuscript technically sound, and do the data support the conclusions?

Reviewer #1: Yes

Reviewer #2: Yes

3. Has the statistical analysis been performed appropriately and rigorously? 

Reviewer #1: Yes

Reviewer #2: Yes

4. Have the authors made all data underlying the findings in their manuscript fully available?

Reviewer #1: Yes

Reviewer #2: Yes

5. Is the manuscript presented in an intelligible fashion and written in standard English?

Reviewer #1: Yes

Reviewer #2: Yes

6. Review Comments to the Author

Reviewer #1: The authors have improved the article and have satisfactorily answered my questions.

The article rigorously analyses the bias in different journals regarding the peer review, analyzing aspects collateral to the context of the submitted manuscript's content.

Reviewer #2: Thank you to the authors for this review and the thorough response to the previous comments. The authors thoroughly explored the research question and developed a clear understanding of the concepts presented. However, some of these concepts might be difficult for a reader encountering the manuscript for the first time. The revision has made some of these concepts easier to understand. All the questions were addressed. The flow of the whole manuscript could be still improved, sometimes being written to complicated and mixed in some concepts. For example, in the introduction section, to postulate the hypothesis the authors stated: "We quantified a possible association at the editorial level between reputation and the outcome of the publication process. We focused on ties between authors, editors and the journal’s impact factor (IF)." which, to me, sounds somewhat "cryptic". I would have preferred a clearer and more straightforward statement, such as: "We hypothesized an association between the number of publications among authors and editors with ties compared to those without ties". After that, the authors should present the objective of the study. However, it seems that what follows is a mix of the aim and the methods used to achieve it. Finally, I believe the manuscript is understandable, and I acknowledge that the writing style might be a matter of personal preference.

7. PLOS authors have the option to publish the peer review history of their article (what does this mean? ). If published, this will include your full peer review and any attached files.

**Do you want your identity to be public for this peer review?** For information about this choice, including consent withdrawal, please see our Privacy Policy .

Reviewer #1: No

Reviewer #2: No

---

## [Author Response · Author response to Decision Letter 2]

15 Dec 2024

Please see our point by point answer file uploaded together with the manuscript

---

## [Decision Letter · Decision Letter 2]

14 Feb 2025

PONE-D-23-31262R2Reputation shortcoming in academic publishingPLOS ONE

Dear Dr. Neveu,

Thank you for submitting your manuscript to PLOS ONE. After careful consideration, we feel that it has merit but does not fully meet PLOS ONE’s publication criteria as it currently stands. Therefore, we invite you to submit a revised version of the manuscript that addresses the points raised during the review process.

We look forward to receiving your revised manuscript.

Kind regards,

Shimpei Miyamoto

Academic Editor

PLOS ONE

Journal Requirements:

Reviewers' comments:

Reviewer's Responses to Questions

**Comments to the Author**

1. If the authors have adequately addressed your comments raised in a previous round of review and you feel that this manuscript is now acceptable for publication, you may indicate that here to bypass the “Comments to the Author” section, enter your conflict of interest statement in the “Confidential to Editor” section, and submit your "Accept" recommendation.

Reviewer #3: All comments have been addressed

Reviewer #4: All comments have been addressed

Reviewer #5: (No Response)

2. Is the manuscript technically sound, and do the data support the conclusions?

Reviewer #3: Yes

Reviewer #4: Yes

Reviewer #5: Yes

3. Has the statistical analysis been performed appropriately and rigorously? 

Reviewer #3: Yes

Reviewer #4: Yes

Reviewer #5: I Don't Know

4. Have the authors made all data underlying the findings in their manuscript fully available?

Reviewer #3: Yes

Reviewer #4: Yes

Reviewer #5: Yes

5. Is the manuscript presented in an intelligible fashion and written in standard English?

Reviewer #3: Yes

Reviewer #4: Yes

Reviewer #5: Yes

6. Review Comments to the Author

Reviewer #3: I have enjoyed reading this interesting work. The authors address a problem that currently exists in relation to professional editors with little academic experience. The results clearly show that their selection as editors is related to the increase in publications by the co-authors of these same editors. An increase much higher than that observed under other situations such as gender. All this points to a clear lack of experience of the professional editors who have been analyzed, which has repercussions on scientific publication in the journals analyzed. Their lesser academic experience is possibly taken advantage of by their co-authors and the same publishers to increase the number of publications by specific authors, the co-authors of these professional editors and the authors of articles that have already been published in the journals analyzed. As a future line of research, the authors could verify the relationship that exists between these same results and those that would be obtained in journals with academic editors who have greater academic and/or professional prestige.

Reviewer #4: The article examines the factors influencing the editor's decision to accept or reject submitted manuscripts. These factors can impact the quantity of scientific output for both the author and the editor. This is an interesting topic that is of significant interest to the scientific community. Nevertheless, the minor revisions mentioned below may help improve the article.

Introduction:

- The introduction is well-structured and provides an overview of the publication process and the role of reputation. With some minor revisions, it will be even stronger. At the end of the introduction, the coherence of the content diminishes, resulting in the reader anticipating a summary of the research objectives, innovations, and contributions.

Methods:

- This section is organized in a manner that narrates the progression of the research. Concerning supplementary methods all procedures are presented in a logical sequence that enables the reader to trace the gradual development and nuances of the study.

Data analyses: Preprocessing step:

- It is stated, “Note that an author may share the affiliation of the editor without being one of his/her former co-authors.” This statement is not understandable to me, at least. Why might an author mention the editor's institutional affiliation without having a contribution to the authorship?

- In “Former editor’s research affiliation as a reputational input to the editor” the comparison was conducted using the main type of t-test (Independent or Paired Samples t-test), which is important to mention.

- In " Interaction with the setup by journals of a non-financial conflicts of interest policy” it is stated “We complemented the linear mixed model involved in the analysis of the difference . . .” It is recommended to provide a rationale for the use of the linear mixed model.

Discussion:

- It is advisable to consider the ethical implications of the research findings in the conclusion.

Final Remarks:

Reputation in the ecosystem of scientific communication is a phenomenon influenced by various factors. Editors’ professional circle and authors’ track record factors influence the process of scientific production. However, this claim cannot be supported by estimation alone and requires further research, which this article has successfully undertaken. It must be acknowledged that managerial positions in higher education institutions can also influence scientific publications. The findings of this research are beneficial for audiences interested in the field of scientific publishing.

Reviewer #5: The paper ‘Reputation shortcoming in academic publishing’ used an example of Nature family journals to detect possible biases in accepting decision made by editors toward manuscripts submitted by certain groups of authors. Specifically, the paper studied former co-authors of editors, authors from former editors’ affiliations, authors with papers in Nature journals, the influence of impact-factors variations of journals. Given that the topic attracts high attention from the perspective of transparency in academic publishing, on the one hand, and poor knowledge on relationship between authors, editors and reviewers, on the other hand, the paper can make significant contribution in our understanding of hidden publishing processes.

As it is seen, the paper has already been reviewed and seems to be balanced, well-structured and detailed. Thus, I do not have many comments. Furthermore, I admire the authors’ comprehensive and thoroughly prepared appendix that can be considered as a good model for other researchers in the processes of describing methodology.

My general comment concerns some difficulties in reading, as many details become clear only with further reading or addressing to the appendix. Thus, perhaps, it will be useful to:

1) explain the notion of “track record” on first mention;

2) explain all abbreviations (e.g., IF, SD…)

3) describe in more detail the recent concept of non-financial conflict of interests. As it was a landmark in achieving higher degree of transparency, I believe the readers will be interested in specific points of its application in academic journals.

One of the conclusions of the paper concerns promotion of triple blind model in peer-review. However, the opposite ‘open identities’ model is currently also being promoted. Thus, it will be interesting to know the authors’ attitude to this model as well. Can it be helpful to eliminate the detected biases in decision-making by editors? By the way, the authors selected the journal with single-blind model for their submission.

7. PLOS authors have the option to publish the peer review history of their article (what does this mean? ). If published, this will include your full peer review and any attached files.

**Do you want your identity to be public for this peer review?** For information about this choice, including consent withdrawal, please see our Privacy Policy .

Reviewer #3: **Yes: ** Jose A. Garcia

Reviewer #4: **Yes: ** Mohammad Reza Ghane

Reviewer #5: No

---

## [Author Response · Author response to Decision Letter 3]

23 Feb 2025

See uploaded file entitled "Neveu et al point by point response after revisions.pdf"

---

## [Editor Report · Decision Letter 3]

17 Mar 2025

Reputation shortcoming in academic publishing

PONE-D-23-31262R3

Dear Dr. Neveu,

We’re pleased to inform you that your manuscript has been judged scientifically suitable for publication and will be formally accepted for publication once it meets all outstanding technical requirements.

Kind regards,

Shimpei Miyamoto

Academic Editor

PLOS ONE
---

## [Editor Report · Acceptance letter]

PONE-D-23-31262R3

PLOS ONE

Dear Dr. Neveu,

I'm pleased to inform you that your manuscript has been deemed suitable for publication in PLOS ONE. Congratulations! Your manuscript is now being handed over to our production team.

Kind regards,

on behalf of

Dr. Shimpei Miyamoto

Academic Editor

PLOS ONE